

# Equifinality Contaminates the Sensitivity Analysis of Process-
# Based Snow Models
Tek Kshetri[1], Amir Khatibi[2], Yiwen Mok[3], M Shahabul Alam[4], Hongli Liu[5], Martyn P. Clark[6]
[1]Water, Sediment, Hazards, & Earth-surface Dynamics (waterSHED) Lab, University of Calgary, Canada
[2]Department of Civil, Geological and Environmental Engineering, University of Saskatchewan, Canada
[3]Department of Civil Engineering, Faculty of Engineering, Universiti Putra Malaysia (UPM), Serdang, Malaysia
[4]Alabama Water Institute, The University of Alabama, USA
[5]Department of Civil and Environmental Engineering, University of Alberta, Canada
[6]Department of Civil Engineering, University of Calgary, Canada
*Correspondence to*: Amir Khatibi (amir.khatibi@usask.ca), Tek Kshetri (iamtekson@email.com)
**Abstract**
This study assesses the impact of different flux parameterizations and model parameters on simulations of snow depth.
Through a sensitivity analysis in a process-based snow model based on the SUMMA framework, various options for
parametrizing snow processes and adjusting parameter values were evaluated to identify optimal modeling
approaches, understand sources of uncertainty, and determine reasons for model weaknesses. The study focused on
model parameterizations of precipitation partitioning, liquid water flow, snow albedo, atmospheric stability, and
thermal conductivity. In this study, sensitivity analysis (SA) is performed using the one-at-a-time (OAT) SA method
as well as the Morris Method to estimate Elementary Effects, aiming to further explore the magnitudes and patterns
of sensitivities. The sensitivity analyses in this study are used to evaluate process parameterizations, model parameter
values, and model configurations. Performance metrics such as the Nash-Sutcliffe Efficiency (NSE), the Kling-Gupta
Efficiency (KGE), the root mean squared log error (RMSLE), and mean are used to assess the similarity between
simulated and observed data. Bootstrapping is employed to estimate the variability of mean Elementary Effects and
establish confidence bounds. The key findings of this research indicate that sensitivity analysis of snow modelling
parameters plays a crucial role in understanding their impact on decision outcomes. The study identified the most
sensitive parameters, such as critical temperature and thermal conductivity of snow, as well as liquid water drainage
parameters. It was observed that water balance fluxes exhibited higher sensitivity than energy balance fluxes in
simulating snow processes. The analysis also highlighted the importance of accurately representing water balance
processes in snow models for improved accuracy and reliability. A key finding in this study is that the sensitivity of
performance metrics to model parameters is contaminated by equifinality (i.e., parameter perturbations lead to similar
performance metrics for quite different snow depth time series), and hence many published parameter sensitivity
studies may provide misleading results. These findings have implications for snow hydrology research and water
resource management, providing valuable insights for optimizing snow modelling and enhancing decision-making.

**Keywords:** Snow Modelling, Sensitivity Analysis, Morris Method, Performance Metrics, Equifinality, SUMMA

**1    Introduction**

Hydrological models differ in their conceptualization and implementation to the extent that there is typically little
agreement regarding "correct" model structures, particularly at larger spatial scales (Clark et al., 2011; Gupta et al.,



2012). Hydrological modelling decisions are often made in an ad-hoc manner, depending on the purpose of the
modelling study, the understanding of hydrological processes, the availability of data for the model evaluation, and
other considerations (Clark et al., 2011). The wide range of decisions made by various modelers to account for the
complex and interdisciplinary challenge of developing process-based hydrological models has led to the proliferation
of hydrological models (Clark et al., 2011; 2015b), making it difficult to clearly attribute model performance
differences to specific decisions that are made as part of the model development process (Koster and Milly, 1997). As
a result, a key challenge for the hydrological modelling community has been to evaluate the underlying hypotheses of
hydrological models, and to further improve models for specific applications (Clark et al., 2011).

The objective of this study is to conduct a sensitivity analysis to evaluate the impact of alternative flux
parameterizations and different model parameters on snow depth in a process-based snow model. To this end, this
study uses the SUMMA framework (described in the next section) to conduct a sensitivity analysis on snow modelling
processes in the Reynolds Mountain East research catchment. An Elementary Effects method is used to evaluate the
options used to parametrize snow modelling processes and the parameter values that are used in the model
parametrizations. This work is useful to identify preferable modelling approaches, understand various sources of
model uncertainty, and determine specific reasons for model weaknesses. The snow process parameterizations we
consider include precipitation partitioning, liquid water flow, snow albedo, atmospheric stability, and thermal
conductivity.
The remainder of this paper is organized as follows. Section 2 describes the methods used to conduct sensitivity
analysis of process-based snow models. Section 3 presents the results and discussion, and Section 4 provides some
key conclusions from this study.
## 2    Methodology
### 2.1    Study Area
Reynolds Mountain East is located in southwestern Idaho (Flerchinger et al., 2012; Reba et al., 2009, 2014, 2011,
2012). It is a headwater catchment in southwestern Idaho's Reynolds Creek Experimental Watershed (RCEW) (Robins
et al., 1965). RCEW's geography varies from a flat valley in the north to steep mountain slopes in the south, with an
elevation range of 1099 to 2244 m above mean sea level and an area of 239 km$^2$ (Sridhar and Nayak, 2010). The
elevation over RCEW causes orographic effects, which cause a decrease in temperature and a rise in precipitation.
The lower elevations of RCEW receive 4-5 times less precipitation than those at higher elevations (Hanson et al.,
2001). At higher elevations, snow predominates, while rain takes over at lower elevations and in the watershed's valley
regions. The Reynolds Mountain East (RME) watershed receives approximately 900 mm of precipitation each year,
with over 70% of it falling as snow, and about 520 mm of this precipitation exits the basin as stream flow (Seyfried et
al., 2009). Snow tends to accumulate in deep drifts, often exceeding 2 meters in sheltered areas (Winstral and Marks,
76   2014).

The data from the "sheltered site" (i.e., the Aspen site) in Reynolds Mountain East (RME) was used to investigate
how simulations of snow depth are affected by the process parametrizations and parameter values, including the
thermal conductivity of snow, snow albedo, and the critical temperature that is used to distinguish rain from snow.
The sheltered site is located in a clearing in a forest, where the grasses are covered by snow early in the snow season
and vegetation has little influence on how the snow depth changes during the year (Figure 1). In addition, protected
locations have modest wind speeds (Reba et al., 2011), thus turbulent heat fluxes have less impact on the surface
energy balance than in more exposed locations. The annual development of the snowpack is mostly governed by the
surface radiation budget. The available observed snow depth data spans from October 1999 to October 2008.





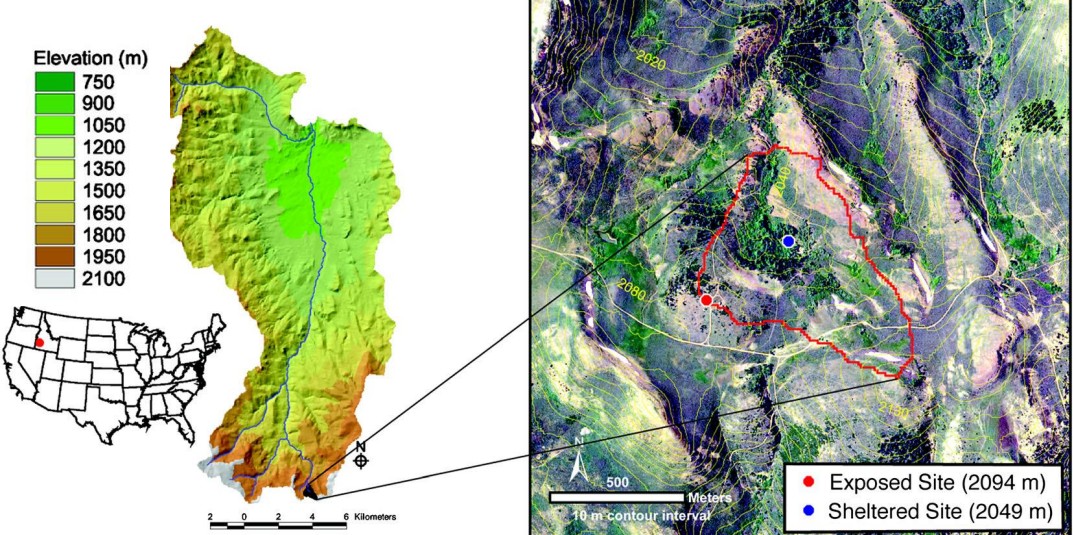

**Figure 1:** Location map of Reynolds Mountain East Catchment (Reba et al. 2012, 2014)

This study simulates snow processes for the period November 2005 to June 2006. This choice was made to reduce the computational effort required for modelling and analysis. Additionally, the time interval between November and June was specifically chosen as it reflects a complete cycle of snow accumulation and melting, with non-zero snow depth values throughout this period. However, selecting the 2005-2006 year does not offer any advantage over choosing other years since the snow depth pattern remains almost consistent throughout the recorded period.

### 2.2    The SUMMA multiple hypothesis framework

A model can be considered as an assemblage of coupled hypotheses that describe dominant hydrological processes (Clark et al., 2011). In this respect, it is important to investigate hypotheses that define individual processes and their connections within the overall system (Clark et al., 2011); i.e., to adopt the "multiple working hypotheses" approach defined by Chamberlin (1890) to investigate multiple rational explanations for the phenomenon being studied. This approach also provides a systematic approach to model evaluation and improvement (Clark et al., 2011). Some examples of multiple hypothesis frameworks are JULES (Best et al., 2011), CLM (Lawrence and Chase, 2007), Noah-MP (Niu et al., 2011), and SUMMA (Clark et al. 2015b).

This study uses the Structure for Unifying Multiple Modeling Alternatives (SUMMA), a flexible, extensible, and modular framework to simulate hydrological processes. SUMMA's modular structure enables incorporating different model representations of physical processes in a common set of conservation equations, which makes it possible to systematically evaluate different parameterizations of the same process and understand the impact of different modelling assumptions on model behavior (Clark et al. 2015b). SUMMA is an example of a multiple hypothesis framework (Clark et al., 2011), which enables users to identify process parameterizations consistent with theoretical expectations and observed data, and to characterize model uncertainty through much more extensive and detailed coverage of the model hypothesis space than typical small ensemble multimodel applications.





**2.3    Snow Modelling Parametrizations and Parameters in SUMMA**
SUMMA includes five key processes for snow modelling; 1. Precipitation partitioning; 2. Liquid water flux through
snowpack; 3. Snow albedo; 4. Atmospheric stability; and 5. Thermal conductivity. The precipitation partitioning and
liquid water in snowpack flux are used in the water balance equations, whereas albedo, atmospheric stability, and
thermal conductivity are used in the energy balance equations. The summary of the parameters are listed in Table 1.
Figure 2 summarizes the water and energy balance fluxes that are calculated in SUMMA. This study employs various
parametrizations of these fluxes for the snowpack simulation, which are detailed below.

## Water balance

**Precipitation**

(Partitioning)

**Rain**   **Snow**

## Energy balance

**Turbulent Fluxes**   **Longwave Radiation**   **Solar Radiation**

(Stability)

(Thermal Conductivity)   (Albedo)

# Snowpack

(Drainage Parameterization)

**Snowpack Outflow**

**Figure 2:** The water and energy balance fluxes used in the SUMMA snow modelling simulations.

Each hydrologic process can be expressed by one or more different parameterizations (i.e., equations or formulas).
Each parameterization includes different numbers of parameters. Table 1 summarizes all the parameters together with
their associated range used for snow modeling, on which sensitivity analysis will be conducted. The detailed
explanation of each parameterization is explained in Appendix A.






**Table 1:** Summary and range of parameters used for sensitivity analysis

| Flux | Parameterizations | Parameters (Unit) | Default value | Min. | Max. |
|---|---|---|---|---|---|
| Precipitation flux | Function of wet-bulb temperature (Marks et al., 2013) | Critical temperature (K) | 273.16 | 272.16 | 274.16 |
| Liquid water in snowpack flux | Gravity drainage (Colbeck, 1976; Colbeck and Anderson, 1982) | Capillary retention (K) | 0.06 | 0.01 | 0.1 |
| | | Hydraulic conductivity of snow (m s$^{-1}$) | 0.015 | 0.005 | 0.05 |
| | | Exponent for meltwater flow (-) | 3.0 | 1.0 | 5.0 |
| Atmospheric stability | Standard (Anderson, 1976) | Critical Richardson Number (-) | 0.2 | 0.1 | 1.0 |
| | Louisinv (Louis, 1979) | Louis79 "b" parameter (-) | 9.4 | 9.2 | 9.6 |
| | MahrtExp (Mahrt, 1987) | Mahrt87 eScale (-) | 1.0 | 0.5 | 2.0 |
| Thermal conductivity | smnv 2000 (Smirnova et al., 2000) | Fixed thermal conductivity (W K$^{-1}$ m$^{-1}$) | 0.35 | 0.10 | 1.00 |
| Albedo | Constant decay (Verseghy, 1991) | Constant albedo decay rate (-) | 1.0d+6 | 1.0d+5 | 5.0d+6 |
| | Variable decay (Yang et al., 1997) | Variable albedo decay rate (-) | 1.0d+6 | 1.0d+5 | 5.0d+6 |
| | | Temperature scale growth (-) | 0.04 | 0.02 | 0.06 |
| | | Albedo soot load (-) | 0.3 | 0.1 | 0.5 |


**2.4    Sensitivity Analysis**
Sensitivity analysis is used to evaluate how the model responds to differences in "input factors" in the model
instantiation. The input factors can be the meteorological forcing data, the model parameters, or the subjective
decisions that are made as part of model development (e.g., the choice of process parameterizations). The model
response to differences in input factors are typically characterized using summary statistics of model behavior (e.g.,
an average model flux) or summary statistics of model performance (e.g., the sum of squared differences between
model simulations and observations).



In this study, sensitivity analysis is performed in SUMMA using local or one-at-a-time (OAT) methods as well as the
Morris Method to estimate Elementary Effects to assess the selection of parameterizations (i.e., equations used to
parameterize specific processes), the selection of model parameters used in the parameterizations (i.e., the model
equations), and the model discretization configurations (Clark et al. 2015b). The sensitivity analysis was completed
using the online collaboration environment operated by the Consortium of Universities for the Advancement of
Hydrologic Science, Incorporated (CUAHSI). CUAHSI provides a cloud computing service called CUAHSI
Community JupyterHub (Tarboton et al., 2024), which includes pre-installed Python wrappers around the SUMMA
model (pySumma).
The following sections provide details of the summary statistics examined as well as the Elementary Effects (EE)
method that is used for sensitivity analysis.

### 2.4.1    The Elementary Effects method (Morris method)

The Elementary Effects (EE) method is a simple yet efficient approach to sifting through numerous input factors in a
model and identifying the significant ones. Morris conceived the concept of elementary effects in 1991, suggesting
the creation of two sensitivity measures to gauge the impact of input factors as negligible, linear and additive, or
nonlinear and intertwined with other factors (Saltelli et al., 2008).
Based on the one-at-a-time method, an individual trajectory is created by perturbing each parameter $p_i$ by a variation
$\Delta_i$. The number of perturbations of each trajectory is equal to the number of input factors ($i = 1,2, \cdots k$). The *EE* of the
$i^{th}$ parameter ($EE_i$) is calculated as follows:
$$EE_i = \frac{f\left(X_{p_i+\Delta_i}\right) - f(X|_{pi})}{\Delta_i}$$

where $f(X)$ denotes the performance metrics used for sensitivity analysis. Here we used four metrics as will be
discussed in the next section. Starting from multiple points within the feasible parameter space, multiple trajectories
($r$) are generated to compute the sensitivity indices, i.e., the mean of *EEs* ($\mu_i^*$) denoting the global sensitivity of each
parameter and the standard deviation of *EEs* ($\sigma_i$) denoting the interaction with other parameters. The equation below
gives the calculations of these indices suggested by Campolongo et al. (2007):
$$\mu_i^* = \frac{1}{r}\sum_{j=1}^{r} \left|EE_i^j\right|, \sigma_i = \sqrt{\frac{1}{r-1}\sum_{j=1}^{r} (EE_i^j - \mu_i^*)^2} \, ,$$

where $EE_i^j$ denotes the $EE_i$ of the $j^{th}$ trajectory.
In summary, to compute each elementary effect, r trajectories of $(k+1)$ points in the input space are required, where $k$
represents the number of input factors. Each trajectory provides $k$ elementary effects, one per input factor, resulting in
$r(k+1)$ sample points in total. An $r = 20$ which is twice more than the typical number of trajectories was selected to
generate the sampling data for each of the parametrizations.
Matrices of sampling data were produced using SAFEpython (Pianosi et al., 2015; Noacco et al., 2019) where the
results were further validated against the principles of the Morris method to ensure the generation of accurate values.
The performance metrics used for the sensitivity analysis are discussed in the next section.

### 2.4.2    Performance metrics

The similarity between the simulated and observed snow depth values over the runtime period was expressed in a
single number using a set of performance/evaluation metrics: The performance metrics used in this study are the Nash-





Sutcliffe Efficiency (NSE; Nash and Sutcliffe, 1970), the Kling-Gupta Efficiency (2012) (KGE; Gupta et al., 2009;
Kling et al., 2012), the root mean squared log error (RMSLE; Willmott and Matsuura, 2005) and the Mean. These
four metrics provide different evaluation perspectives of flow simulation results, and each will identify in a distinct
manner the extent to which altering a parameter influences the outcome of a decision. Accordingly, using these metrics
together with qualitative sensitivity analysis methods will help identify model configurations that closely predict the
overall system behavior.
Table 2 defines the four performance metrics, where n is the total number of events; $O_i$ and $S_i$ are the observed data
and simulated data; $\underline{O}$ is the corresponding mean values; $\sigma_o$ and $\sigma_s$ represent the standard deviation of the observed
and simulated values; $\mu_o$ and $\mu_s$ represent the mathematical expectation of the observed and simulated values,
respectively.
**Table 2:** Description of the performance metrics

| Metrics | Equation | Scope |
|---|---|---|
| Nash-Sutcliffe Efficiency (NSE) | $$NSE = 1 - \frac{\sum_{i=1}^{n} (S_i - O_i)^2}{\sum_{i=1}^{n} (O_i - \underline{O})^2}$$ | -inf < NSE < 1 |
| Kling-Gupta Efficiency 2012 (KGE) | $$KGE_{2012} = 1 - ED$$ $$ED = \sqrt{(s[1] \cdot (r - 1))^2 + (s[2] \cdot (\gamma - 1))^2 + (s[3] \cdot (\beta - 1))^2}$$ *s (tuple of length three) = Represents the scaling factors to be used for re-scaling of the coefficients.* *r = Pearson Correlation Coefficient* $$\beta = \mu_s/\mu_o$$ $$\gamma = \frac{CV_s}{CV_o} = \frac{\sigma_s/\mu_s}{\sigma_o/\mu_o}$$ | -inf < KGE (2012) < 1 |
| Root Mean Squared Log Error (RMSLE) | $$RMSLE = \left(\frac{1}{n}\sum_{i=0}^{n} \left(ln\ ln\ \left(\frac{S_i}{O_i}\right)\right)^2\right)^{\frac{1}{2}}$$ | 0 ≤ RMSLE ≤ inf |
| Mean (average snow depth) | $$Mean = \frac{1}{n}\sum_{i=0}^{n} (S_i)$$ | 0 ≤ Mean < inf |

The NSE is a commonly used metric that normalizes model performance into an interpretable scale (Knoben et al.,
2019). One main drawback of the NSE is that it punishes a higher variance in the observed values (Roberts et al.,
2018). Gupta et al. (2009) developed KGE (2009) in the context of hydrologic modelling to explain the relative
significance of correlation, bias, and variability to address issues associated with the NSE. Further, Kling et al. (2012)
proposed KGE (2012) as an enhanced version of KGE (2009) in order to prevent cross-correlation between variability
ratios and bias. Larger values of NSE and KGE indicate a stronger agreement between observations and simulations.
When the metric values for NSE, Mean, and RMSLE are all zero, or for KGE when the value is -0.41, it indicates that
the model simulations have the same explanatory power as the mean of the observations (Knoben et al., 2019). The
RMSLE limits the impact of outliers by more evenly weighting high and low values. Smaller values indicate a stronger
correspondence between observations and simulations (Roberts et al., 2018).
Performance metrics were calculated utilizing the HydroErr Python package (Roberts et al., 2018). The metrics were
computed for each simulation run, with the number of simulations for each parametrization being *r(k+1)*, as described



in section 3.3.1. This enabled the calculation of Elementary Effects for each parametrization. Bootstrapping was used
as a resampling technique to estimate the variability of mean Elementary Effects and to develop confidence bounds.
The method involved randomly selecting a sample from the original data set and generating multiple resampled data
sets of the same size as the original data set. Then, the mean Elementary Effects were calculated for each resampled
data set, resulting in a distribution of the means. Confidence bounds were then determined by calculating the 25th,
50th, and 75th percentiles.

## 3    Results and Discussion

In this section, we outline our systematic approach to analyzing parameter sensitivities within our snow modeling
study. We begin by utilizing SUMMA to conduct simulations under varying parameter perturbations. These
preliminary simulations enable identifying potentially sensitive parameters, offering initial insights into their impact
on snow depth predictions. Subsequently, we quantitatively assess sensitivities using sensitivity indices derived from
diverse performance metrics (Mean, KGE, MSE, and RMSLE). This comprehensive analysis highlights the agreement
between simulations and observations across different parameterizations, revealing the potential influence of specific
parameters on specific performance metrics. Additionally, we employ the Morris Method to estimate Elementary
Effects, aiming to further explore the magnitudes and patterns of sensitivities. This method aids in characterizing
parameter impacts and contributes to a deeper understanding of the complex interplay between parameters and
simulation results. Overall, our approach enables a comprehensive examination of parameter sensitivities based on
specific performance metrics, providing specific parameters influence on the snow model.

### 3.1 Snow depth perturbation

SUMMA was used to run the parameters listed in Table 1 by altering one parameter at a time while maintaining a
default value for other parameters. Figure 4 illustrates the result of these simulations for 12 parameters and
parametrizations used for snow modelling in the Reynolds Mountain East research catchment. These snow depth
perturbation graphs are helpful to gain an initial understanding of potentially high and low sensitivity parameters.
Based on these initial sensitivity experiments, it is expected that the most sensitive parameter would be the critical
temperature and thermal conductivity of snow (Smnv2000 method), followed by the liquid water drainage parameters
(capillary retention and exponent of meltwater flow), and the critical Richardson Number (Standard atmospheric
stability method), respectively. Furthermore, it can be observed that snow depth calculation is relatively insensitive to
changes in snow albedo, as well as to the atmospheric stability methods of MahrtExp and Louisinv.





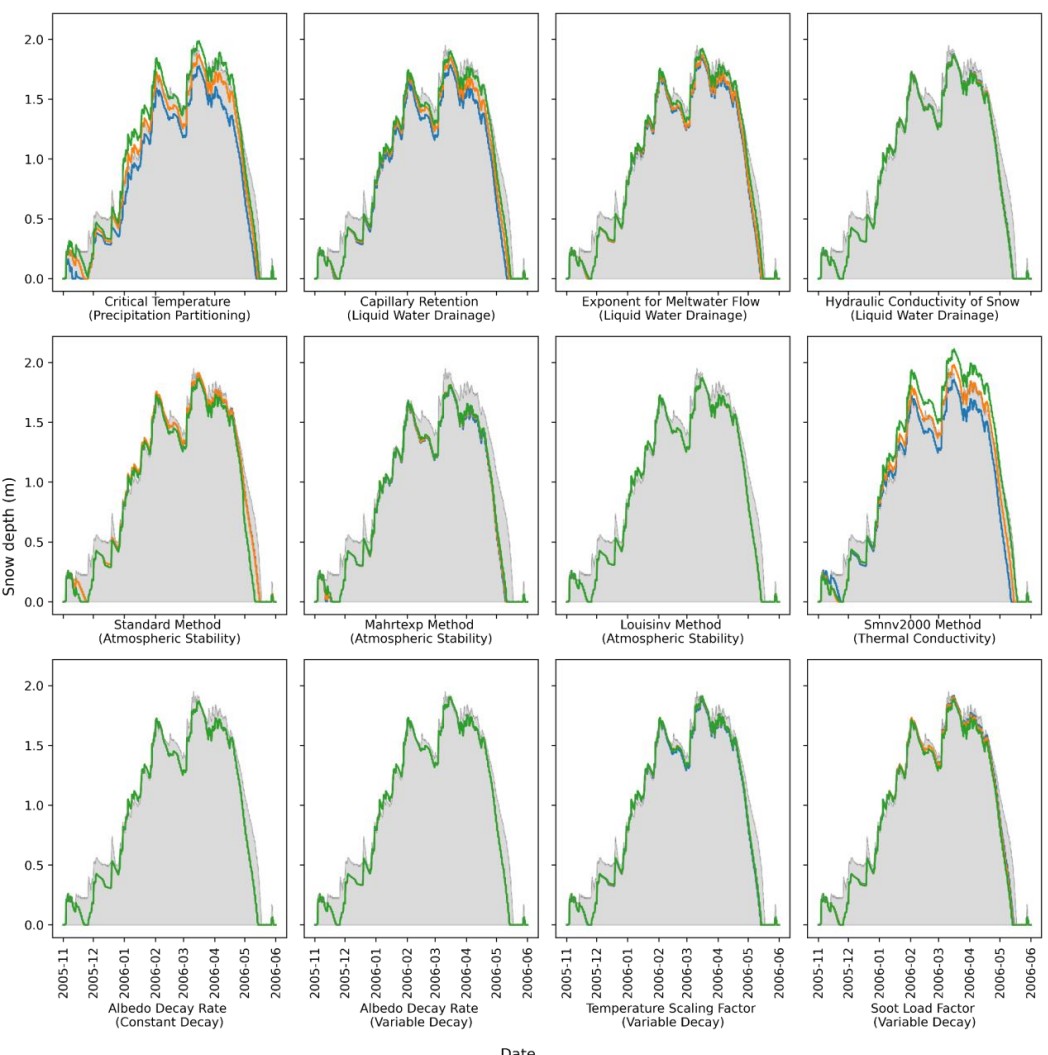

**Figure 3:** Snow depth perturbation plots for all parameters and available parameterization used in snow modelling processes (November 2005 to June 2006). The gray areas represent observed snow depth values, while the blue, orange, and green lines represent simulated snow depth values for the minimum, default, and maximum values of a parameter, respectively.

### 3.2 Confidence intervals and marginal errors

To quantify the magnitude and rank of sensitivity associated with each parameter, a more comprehensive One-at-A-Time sensitivity analysis is conducted to generate sensitivity indices for every parameter. Table 4 summarizes the mean of the 95% confidence interval ± marginal error of the performance metrics. This helps to determine the result's precision and reliability and to assess the statistical significance of the results. The margin of error for four



performance metrics is relatively small, indicating a more precise estimate.

The reported values of Mean snow depth, NSE, KGE (2012), and RMSLE performance metrics provide a measure of
how well the simulations using each parameterization capture the observed snow depth values. Overall, the
performance results show a high degree of consistency between the simulated and observed snow depth data based on
mean metric, regardless of the parametrizations. However, ranking of each parameter's sensitivity from least to most
accurate slightly differs depending on the performance metric used. In general, the results suggest that the Louisinv
parameterization/method is capable of providing more accurate predictions for estimating atmospheric stability as
compared to the MahrtEXP and Standard methods, which show lower Mean, NSE, and KGE scores and a higher
RMSLE score. The relatively higher accuracy of Louisinv is likely due to the iterative nature of the Louisinv method,
which allows for a more precise calculation of the bulk Richardson number and eddy diffusivities for heat and
moisture. It can also be observed that the accuracy of predictions is almost identical for constant and variable albedo
decay rates. This suggests that the temporal changes in snow properties used for albedo calculation can be adequately
captured through the accumulation and melting period decay curves at Reynolds Mountain East. Overall, the
MahrtExp method is associated with the least accurate prediction, while the thermal conductivity and variable albedo
decay parametrization are associated with the highest prediction accuracy.

**Table 4:** Performance metrics associated with each parametrization

| Flux | Parameterization | Mean | KGE (2012) | NSE | RMSLE |
|------|------------------|------|------------|-----|-------|
| Precipitation flux | Function of wet-bulb temperature (Marks et al., 2013) | 0.933 ± 0.017 | 0.830 ± 0.017 | 0.939 ± 0.008 | 0.097 ± 0.005 |
| Liquid water in snowpack flux | Gravity drainage (Colbeck, 1976; Colbeck and Anderson, 1982) | 0.916 ± 0.008 | 0.819 ± 0.008 | 0.933 ± 0.008 | 0.101 ± 0.005 |
| Atmospheric stability | Standard (Anderson, 1976) | 0.931 ± 0.005 | 0.819 ± 0.010 | 0.940 ± 0.006 | 0.102 ± 0.005 |
| | Louisinv (Louis, 1979) | 0.924 ± 0.000 | 0.826 ± 0.000 | 0.944 ± 0.000 | 0.098 ± 0.000 |
| | MahrtExp (Mahrt, 1987) | 0.861 ± 0.002 | 0.747 ± 0.003 | 0.880 ± 0.002 | 0.137 ± 0.001 |
| Thermal conductivity | smnv 2000 (Smirnova et al., 2000) | 1.023 ± 0.015 | 0.850 ± 0.004 | 0.949 ± 0.003 | 0.085 ± 0.003 |
| Albedo | Constant decay (Verseghy, 1991) | 0.933 ± 0.002 | 0.834 ± 0.002 | 0.949 ± 0.001 | 0.094 ± 0.001 |
| | Variable decay (Yang et al., 1997) | 0.938 ± 0.002 | 0.835 ± 0.002 | 0.952 ± 0.001 | 0.093 ± 0.001 |



**3.3 Qualitative sensitivity analysis**
Figure 5 shows a scatter plot of all snow model parameters, where the mean snow depth is plotted against each
parameter range. The purpose of this plot is to qualitatively visualize parameters with potentially high and low
sensitivities, which can be useful for screening and ranking them. Parameters with higher variability are considered to




be more sensitive. Based on the observed patterns, it is expected that the constant and variable albedo parameters, as
well as the hydraulic conductivity of the snowpack, will be the least sensitive. On the other hand, the critical
temperature, thermal conductivity, capillary retention, and exponent for meltwater flow are likely to be the most
sensitive parameters. Moreover, in cases where there is a discernible pattern in each parameter, it becomes possible to
identify the optimum value that can lead to the highest level of agreement between the simulation and observation.
For instance, the simulation of snow depth will improve for the values of the exponent for meltwater flow greater than
2, but will not significantly improve for the thermal conductivity values greater than 0.8 [$W\ K^{-1}\ m^{-1}$].

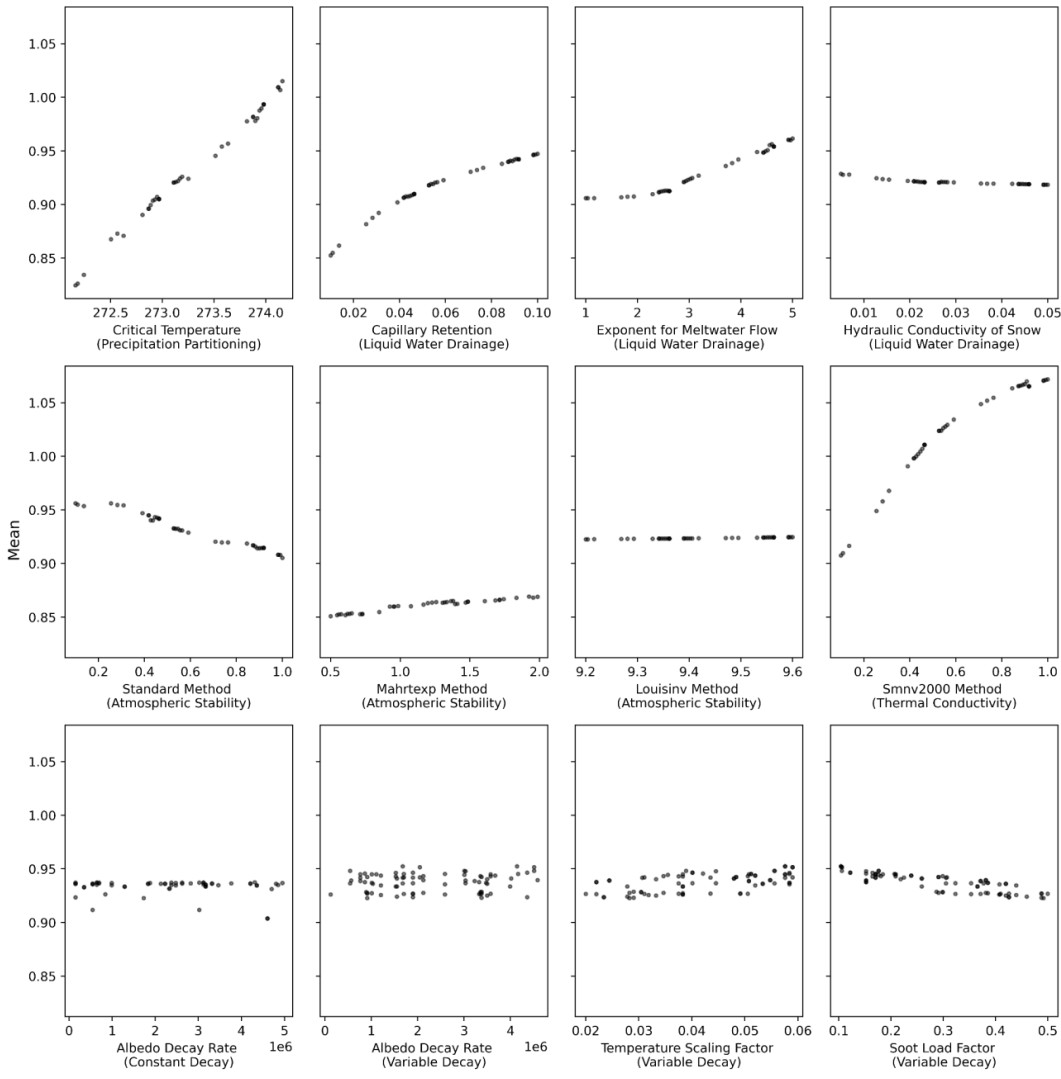

**Figure 4**: Qualitative sensitivity of parameters and parametrizations used in snow modelling process. Scatter plot of
the Mean as the performance metric against parameter ranges.



### 3.4 Quantitative sensitivity analysis

To further examine the accuracy of these qualitative findings, a quantitative sensitivity analysis was employed as discussed in section 3.2.2. The Morris Method was used to estimate the Elementary Effects (EEs) for each of the model parameters. The variance in the ranking of parameters' accuracy prediction by different performance metrics prompted us to develop Elementary Effect functions using all the metrics and compare the sensitivity results. This was done to identify a system-scale performance metric that can consistently describe the observed sensitivities through scattered and perturbation plots.

A bar plot depicting the mean of Elementary Effects was developed accordingly and is shown in Figure 6. The plot shows a sensitivity analysis of all parameters and parametrizations using four different performance metrics. The parameters are sorted in descending sensitivity order, from the highest to the lowest, as determined by the mean snow depth. At a first glance, it can be seen that the performance metrics, which were predicting the accuracy of simulations relatively closely, exhibit substantially different degrees of sensitivity for each parameter.

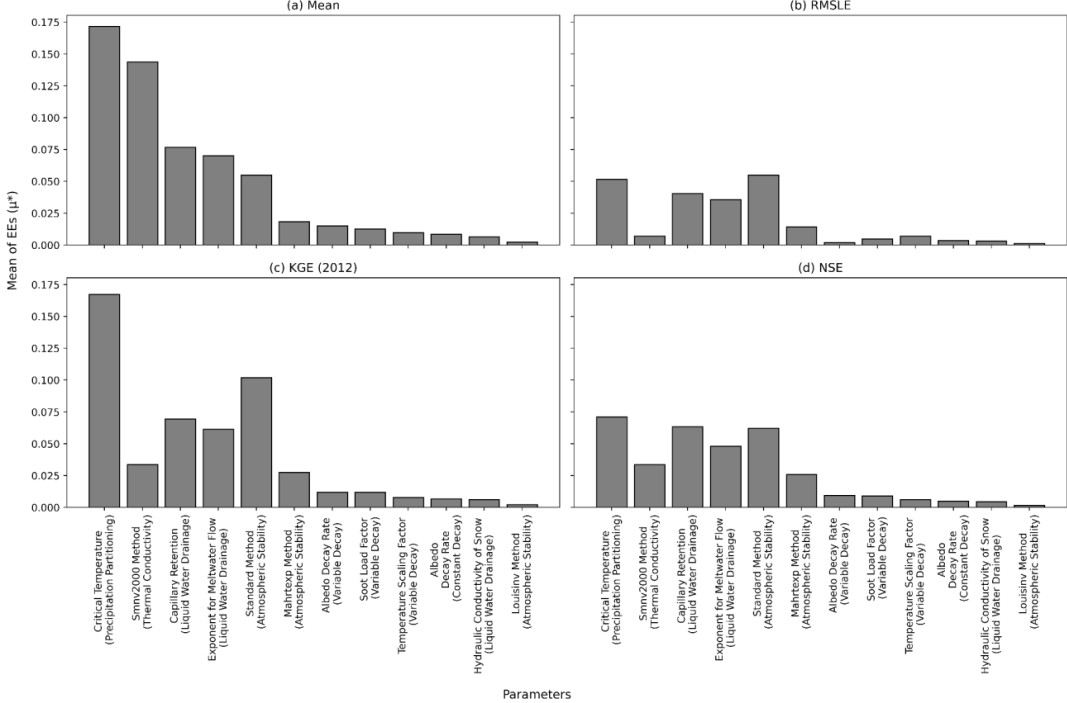

**Figure 5**: Mean of Elementary Effects estimated using four performance metrics; (a) Mean, (b) RMSLE (c) KGE, and (d) NSE.

It can be observed that the critical temperature for rainfall, which determines the partitioning of incoming precipitation into rain and snow, is the most sensitive parameter in determining the depth of the snowpack. This prediction is consistent with previous observations made through perturbation plots in Figures 3 and 4, where a significant fluctuation in snow depth was observed by changing the critical temperature within its range. It was interesting to note that regardless of the performance metrics chosen, parameters such as the hydraulic conductivity of snow, and the constant and variable albedo decay rates, were consistently identified as the least sensitive in snow process modeling. This observation leads to the conclusion that different performance metrics can be effective in determining the



sensitivity indices of the least sensitive parameters. The discrepancy lies in quantifying the magnitude and rank of
parameters with relatively higher sensitivities.
A very interesting result in Figure 5 is the difference in sensitivity for parameters in the thermal conductivity
parameterization for the performance metrics that quantify differences in the mean (top left subplot in Figure 5) from
the performance metrics that quantify differences in the simulated and observed time series (i.e., the RMSLE, KGE,
and NSE, shown in the top-right, bottom-left, and bottom right subplots of Figure 5). These differences in model
sensitivity can be explained through inspection of Figure 3, where the differences between the simulations with lowest
thermal conductivity (blue line) have differences with the observations that are of opposing sign to the model
simulations with the highest thermal conductivity (green line). Importantly, even though the snow depth time series
are quite different, the performance metrics are quite similar. These results are a manifestation of equifinality, where
different model trajectories can have similar performance. In this case, the impact of thermal conductivity parameters
on the time series performance metrics are much smaller than would be expected from visual inspection of the impacts
of thermal conductivity parameters on the snow depth time series.
To further quantify the rank and magnitude of the sensitivities, the means of Elementary Effects were bootstrapped to
create a box and whisker plot, as shown in Figure 6. The parameters in the box and whisker plot are sorted to display
the water balance fluxes (i.e., precipitation partitioning and liquid water flow) first, followed by the energy balance
fluxes (i.e., atmospheric stability, thermal conductivity, and albedo).
Overall, it is noticeable that water balance fluxes exhibit higher sensitivity than energy balance in simulating snow
processes.

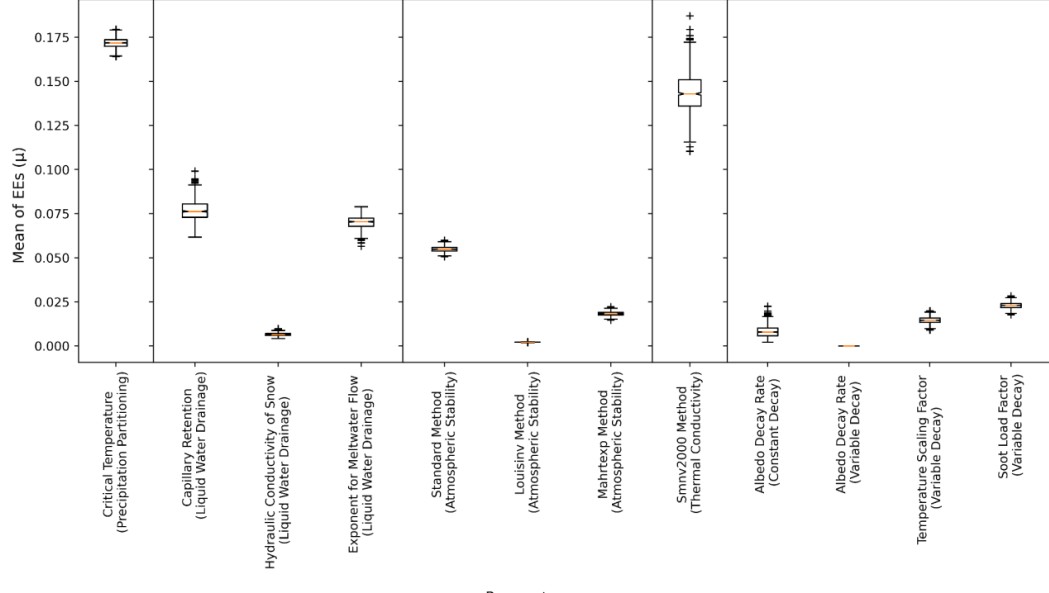

**Figure 6**: Mean of Elementary Effects developed using the Mean performance metric. The boxes show the
interquartile range (25th, 50th, and 75th percentiles) and the whiskers represent the maximum and minimum limits of
the mean of EEs.
Accordingly, the critical temperature of rain is the most sensitive parameter among the water balance parameters,
while the thermal conductivity of snowpack is the most sensitive parameter among the energy balance parameters. As



expected, the albedo parameters, whether constant or variable, are among the least sensitive components in the snow
modelling processes. Figure 6 also shows that the parameters associated with the drainage of liquid from the
snowpack, namely capillary retention, and exponent of meltwater flow, have almost equally high sensitivities. In
contrast, the hydraulic conductivity of the snowpack seems to be among the least influential parameters in determining
the depth of snow. The insensitivity of the latter parameter is in alignment with the observations from perturbation
(Figure 4) and scattered plots (Figure 5).
The third panel of the boxplots in Figure 6 compares the sensitivity of three different atmospheric stability
parametrizations, showing that the critical Richardson number (Standard method) is the most sensitive among the
three methods. The reason for the high sensitivity of the Standard method may be attributed to the Richardson number
being in the numerator with an exponential power, as opposed to the other two methods, where the atmospheric
stability has an inverse relationship with the Richardson number. This sensitivity analysis denotes that water balance
parametrizations (i.e., critical temperature of rain) are much more sensitive than the energy balance (i.e., variable
albedo decay rate), as demonstrated by the comparison of the highest and lowest ends of flux spectrums.

The study's new insights suggest that water balance parameterizations are more sensitive than energy balance fluxes.
This finding highlights the importance of accurately representing the water balance processes in snow models, which
can improve the accuracy of snowmelt predictions. Additionally, the study suggests that modular and flexible
frameworks such as SUMMA enable identifying and isolating sensitive parameters, thereby improving the sensitivity
analysis of snow models.

## 4    Conclusion

This work presents the results of a sensitivity study of model simulations of snow processes in the Reynolds Mountain
East research catchment using a flexible, extensible, and modular hydrological framework. Through the analysis
conducted in this study, the following conclusions can be drawn:

● The use of a modular and flexible framework enables identifying and isolating parameters and parametrizations,
which in turn enables comprehensive sensitivity analysis.
● The sensitivity of performance metrics to perturbations in model parameters is contaminated by equifinality. We
illustrate some cases in this paper where parameter perturbations lead to similar performance metrics for quite
different snow depth time series. Given that many published parameter sensitivity studies are based on the
sensitivity of performance metrics to model parameters, the conclusions from many model sensitivity analysis
studies may not be trustworthy. It is hence crucial to select metrics for sensitivity analysis that accurately represent
the system-scale behavior. This would improve quantifying the magnitude and ranking of sensitivity indices.
● In the specific case of snow modeling in the Reynolds Mountain East research catchment, water balance fluxes are
generally more sensitive than energy balance fluxes. Among the parameters, the critical temperature of rain is the
most sensitive, while the albedo decay rate is the least sensitive.

There are several limitations of this study that can be explored in future research. This study used Elementary Effects
to conduct the sensitivity analysis. Although the Elementary Effect method is useful for models with a large number
of uncertainty factors, it does not quantify the relative importance of inputs. Additionally, the sensitivity analysis was
performed by isolating each snow modeling parameter and analyzing its effect on the system individually. This
approach limited the potential interactions between fluxes and may have affected the sensitivity rank and magnitude
of the parameters. Application of methods like variance-based sensitivity analysis methods can address these
limitations.




**Acknowledgement**

The authors acknowledge the cloud computational support from the Consortium of Universities for the Advancement of Hydrologic Science, Incorporated (CUAHSI).

**Code and data availability**

The code and data used for this study can be obtained by directly contacting the corresponding author.

**Author contributions**

AK, MSA, TK, and YM designed the study and carried out the analysis. AK, MSA, TK, and YM wrote the manuscript, with contributions and reviews from MPC and HL. All the authors contributed to the discussion and revision of the manuscript.

**Competing interests**

The authors declare that they have no conflicts of interest.

**Appendix A**

4.1.1    Precipitation Partitioning

The parameterization to partition precipitation into rain and snow is a function of wet-bulb temperature (Clark et al., 2015c; Marks et al., 2013), parametrized as a linear function that describes the temporal variability of the wetbulb temperature over a model time step. The minimum and maximum wetbulb temperature over a model time step are defined as

$$T_{max} = T_{wet} + T_{range}/2$$
$$T_{min} = T_{wet} - T_{range}/2$$

where $T_{wet}$ is the wetbulb temperature and $T_{range}$ defines the temporal variability of the wetbulb temperature over a model time step. The fraction of precipitation that falls as rain, $f_{rain}$, is then defined based on a critical value of the wetbulb temperature, $T_{crit}$, as

$$f_{rain} = \{0 \; T_{max} < T_{crit} \; \frac{T_{max} - T_{crit}}{T_{max} - T_{min}} \; T_{min} \leq T_{crit} \leq 1 \; T_{min} > T_{crit} \; T_{max}$$

and the fraction of precipitation that falls as snow, $f_{snow} = 1 - f_{rain}$. The default value for $T_{crit}$ is 0°C.

4.1.2    Liquid Water Flow

The storage and transmission of liquid water in the snowpack in SUMMA is parameterized as gravity drainage based on the (Colbeck, 1976; Colbeck and Anderson, 1982) and calculated as follows:

$$q = k \left( \frac{\theta_{liq} - \theta_{res}}{\emptyset - \theta_{res}} \right)^c$$

where $q$ is the drainage flux, $k$ is the conductivity of snowpack [$m \; s^{-1}$], $\theta_{liq}$ is the current volumetric water content [-], $\theta_{res}$ is the residual volumetric water content [-], $\emptyset$ is the available fraction of pore spaces [-], and $c$ is non-linearity coefficient [-]. As such, the liquid water flow in snow is controlled by $k, \theta_{res}, c$.





### 4.1.3 Snow Albedo

Two semi-empirical options are implemented in SUMMA for the snow albedo parameterization:

- Variable Decay: This option is derived from Biosphere-Atmosphere Transfer Scheme (BATS) described by Yang et al. (1997) where the albedo decay rate varies as snow properties change over time. BATS represents the albedo separately for visible and near-infrared wavelengths as different wavelengths of solar radiation are absorbed and reflected differently by the snowpack. The direct-beam albedo in BATS is set equal to the diffuse albedo plus an additive factor at high solar zenith angles meaning that the diffuse albedo is the average reflectivity of the snow surface over all angles of incidence, while the additive factor represents the additional reflectivity of the snow surface at low solar zenith angles (Clark et al 2015a, 2015c; Yang et al., 1997). The variable decay albedo parameterization is given as

$$\frac{d\alpha}{dt} = \kappa\alpha$$

$$\kappa = \kappa_0 \ (r_1 + r_2 + r_3)$$

where, $\alpha$ is the snow surface albedo, $\kappa_0$ is the time delay scaling factor, $r_1$ represents effects of grain growth due to vapor diffusion, $r_2$ represents the additional effects of grain growth when the snow temperature is near the freezing point, and $r_3$ is an adjustable parameter representing effects of dirt and soot.

- Constant Decay: This option is derived from the Canadian Land Surface Scheme (CLASS) described by Verseghy (1991) where the albedo decay rate, which controls the rate at which the snow albedo adjusts to changes in snow properties, is temporally constant. CLASS uses two decay curves, one for periods of accumulation and the other for melting periods which is because the physical processes that affect snow albedo, such as grain growth and meltwater formation, are different during these two periods. This approach assumes that the direct beam and diffusive albedos are identical, as the difference between the two is only distinct at high solar zenith angles when shortwave radiation fluxes are small (Clark et al., 2015a, 2015c; Verseghy, 1991). The constant decay albedo parameterization is given simply as

$$\kappa = \kappa_0$$

### 4.1.4 Atmospheric Stability

A considerable amount of the energy involved in snowmelt or ice ablation is transferred from the atmosphere through turbulent fluxes (see Figure 2). The turbulent transfer process in the surface boundary layer generates sensible and latent heat fluxes, which are recognized as important contributors to the energy input for melting snow cover (Morris, 1989). These fluxes can be expressed as the covariance of fluctuations in vertical velocity. While it is feasible to directly measure these covariances and obtain accurate estimates of the fluxes, such measurements are seldom accessible. In numerous experimental and modeling studies, it becomes essential to represent these fluxes through parameterization. Three options are implemented in SUMMA for the atmospheric stability parametrization based on the bulk Richardson number:

- Critical Richardson Number (Standard): This option is the default atmospheric stability option in SUMMA and derived from Anderson (1976) which calculates atmospheric stability based on the vertical gradient of potential temperature and the horizontal wind speeds. The bulk Richardson number (Ri) is defined as the ratio of the potential energy available for turbulence to the kinetic energy associated with the vertical shear of the horizontal wind. This method assumes that the eddy diffusivity for heat is constant and depends only on the bulk Richardson number. Implementing this method in SUMMA is based on the use of atmospheric stability classes, which are assigned based on the value of the bulk Richardson number (Anderson, 1976; Clark et al. 2015a, 2015c;). The atmospheric stability correction (F) for the stable condition can be calculated



by using the following formula:

$$F = \{(1 - 5R_i)^2 \quad R_i < R_{ic} \quad 0 \qquad R_i \geq R_{ic}$$

where, $R_i$ is the bulk Richardson Number [-].

● Louis79 "b" parameter (Louisinv): This option is derived from Louis (1979), who parameterizes the
atmospheric stability correction (F) for stable conditions as:
$$F = \frac{1}{(1 + b'R_i)^2}$$
where, $b'$ = Louis (1979) "b" parameter / 2 [-].

● Mahrt87 eScale (MahrtExp): This option is derived from Mahrt (1987), who considers spatial averaging of
subgrid processes. where the atmospheric stability correction ($F$) for stable conditions is given as:
$$F = e^{(-m\,R_i)}$$
where, $m$ = Mahrt exp coefficient [-].

**4.1.5   Thermal Conductivity**
Four empirical options are implemented in SUMMA to parameterize the thermal conductivity of snow:

tyen 1965: This option is derived from Yen (1965) who conducted experiments to measure the effective thermal
conductivity ($K$) of naturally compacted snow at different densities. The results of Yen's experiments showed that the
effective thermal conductivity of ventilated snow strongly depends on the square of its density plus the effect of dry
airflow. Yen concluded that turbulence introduced by the air stream reduces the resistance for the simultaneous transfer
processes of heat and mass (i.e., water vapor diffusion) and increases the thermal conductivity of snow. Accordingly,
as the density of snow and mass flow rate of dry air increases, the effective thermal conductivity also increases. In the
absence of airflow, Yen's equation reduces to a relationship between the effective thermal conductivity and the snow
density.

$$K = 3.217 \times 10^{-6}\,\gamma_s{}^2$$

where, $\gamma_s$ is the bulk density of snow [$kg.m^{-3}$].

● melr 1977: This option is derived from Mellor (1977) who described heat transfer in a dry snowpack as a
process involving conduction (e.g., in the network of ice grains and bonds, and airspaces and pores),
convection and radiation (e.g., across pores which were assumed negligible), and water vapor diffusion
through voids; however, considering empirical curves of the snow thermal conductivity developed by a
number of researchers, he concluded that engineering applications the thermal conductivity of the dense snow
and bubbly ice can be assumed to be proportional to the square of the snow density (discounting finer details
or complications). Accordingly, SUMMA adopted a fit to the quadratic snow density versus thermal
conductivity data provided by Mellor (Mellor 1977; Clark et al. 2015a, 2015c).

$$K = 2.576 \times 10^{-6}\,\gamma_s{}^2 + 7.4 \times 10^2$$

● jrdn 1991: This option is derived from Jordan (1991) who proposed to estimate the effective thermal
conductivity of snow by accounting for ice and air conduction and vapour diffusion through the voids.



Accordingly, Jordan's parametrization includes a thermal conductivity term with a polynomial relationship
to the snow density together with constant values for the conductivity of air and ice in the snowpack.
Adjustable parameters in Jordan's equation have been selected so that the effective thermal conductivity fits
the data of Yen (1962) and extrapolates to ice conductivity when the snow density is that of ice.
$$K = K_a + (7.75 \times 10^{-5}\,\gamma_s + 1.105 \times 10^{-6}\,\gamma_s^{\,2})(K_i - K_a)$$

where, $K_a$ is thermal conductivity of air = 0.023 [$W\,K^{-1}\,m^{-1}$] and $K_i$ is thermal conductivity of ice = 2.29 [$W\,K^{-1}\,m^{-1}$].
● smnv 2000: This option is derived from Smirnova et al. (2000) where a constant value for the thermal
conductivity of snow (set equal to 0.35 $W\,K^{-1}\,m^{-1}$) was considered which lies within the range of this variable
for old and new snow as per a study.



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
