# Peer review of "Equifinality Contaminates the Sensitivity Analysis of Process Based Snow Models"

_EGUsphere, 2023_

## Author Comment (AC1)

**RC2: Francesca Pianosi**

The paper presents a sensitivity analysis of a process-based snow model looking at how different choices of model parameter values and/or equations to represent snow processes impact snow depth simulations and accuracy with respect to observations available at a site in southwestern Idaho.

Overall, the manuscript is well written though some key aspects of the methodology applied need to be clarified (see point 2 and 3 below). Also, the contribution of the manuscript needs to be better articulated (point 1). I am not an expert of snow process modelling but, as an expert in sensitivity analysis of hydrological models more broadly, I do not find the key finding of the manuscript particularly novel or unexpected, so I think the authors should better articulate the specific contribution of their study and how it will help and inform other process-based snow modellers.

Thank you very much for your thorough review and insightful comments on our manuscript titled "Equifinality Contaminates the Sensitivity Analysis of Process-Based Snow Models." We appreciate your positive feedback on the organization and writing of our article, and we are grateful for your suggestions to improve our manuscript. Below, we address your general comments and minor suggestions in detail. Please find our response in **Blue** color.

**1) TITLE AND CONTRIBUTION**

Starting from the title, the authors highlight as key contribution of their work that sensitivity analysis results "are contaminated" by equifinality, however it is not totally clear to me what this "contamination" means and what its implications are.

**Response:** Thanks for the comment. The title will be updated to better reflect the paper's objective and contribution as has been discussed in response to the following comment.

On L. 356 they say:

"the sensitivity of performance metrics to perturbations in model parameters is contaminated by equifinality. We illustrate some cases in this paper where parameter perturbations lead to similar performance metrics for quite different snow depth time series. Given that many published parameter sensitivity studies are based on the sensitivity of performance metrics to model parameters, the conclusions from many model sensitivity analysis studies may not be trustworthy."

Now the fact that different parameter combinations lead to different simulated time series all associated with similar value of performance metric is not surprising - this is the very definition of equifinality. As for SA, even if a performance metric exhibits small variability, it can be subject to sensitivity analysis: in fact, as shown in Figure 5, also in this work it was possible to clearly estimate the relative importance of parameters in determining the (small) variability of the 3 performance metrics (RMSLE, KGE, NSE). The Figure shows that these sensitivities are a bit different – for example thermal conductivity has low sensitivity index for RMSLE and much higher for KGE, NSE – but again this is not surprising, as we know from SA literature that different metrics are sensitive to different parameters (e.g. see review in:

https://doi.org/10.1016/j.earscirev.2019.04.006) which is also why it is good practice to use a range of metrics for model calibration and evaluation.

To be honest, in this case these differences do not look too dramatic either: while the parameter ranking is not exactly the same across the four panels in Figure 5, the screening results is not significantly changed (the first six parameters are important while the other six are uninfluential) and is also consistent with conclusion from visual inspection of simulated time series in Figure 3. So, I think the authors need to better articulate what they mean by the statement that "equifinality contaminates sensitivity analysis" and in what way "the conclusions from many model sensitivity analysis studies may not be trustworthy". Maybe some specific examples from the snow modelling literature and previous SA applications to snow models would help to make the case of what are the new lessons learnt here.

**Response:** This is a very constructive comment. We believe that to properly address this comment, it needs to be seen together with comment #3. Accordingly, we will distinguish between the "Performance Metrics" (i.e., RMSLE, KGE, NSE) and the "Signature Metric" (i.e., Mean). A signature metric being a quantitative measure that captures key characteristics or patterns of a hydrological time series (McMillan 2020), which essentially can represent the "Model Behavior". Accordingly, the methodology and discussion sections will be improved to articulate as to how the performance metrics mask the model sensitivity to different parameters due to equifinality. In contrast, signature metrics, such as Mean, can describe one of the system behaviors with more fidelity as shown through the consistency between the model simulations and observations.
Further references to previous SA applications to snow models will also be provided to back up this discussion. Additionally, the sentence within which we mention "the conclusions from many model sensitivity analysis studies may not be trustworthy" will be modified to be less subjective.

**2) MODEL AND EXPERIMENTAL SET-UP**

The set-up of the sensitivity analysis and of the model itself needs to be clarified.

On line 143 the authors say that they will assess "the selection of parameterizations (i.e., equations used to parameterize specific processes), the selection of model parameters used in the parameterizations (i.e., the model equations), and the model discretization configurations". However, the "discretization configurations" are not mentioned or reported ever again, and the "selection of parameterizations" is only applied to two processes (atmospheric stability and albedo) out of five modelled (see Table 1). So, this suggests to me that this is mostly a "conventional" sensitivity analysis of model parameters, with the additional assessment of different parameterisations for some of the modelled processes. If so, this needs to be clarified throughout the manuscript.

**Response:** This section will be modified to clarify that there are only two parametrizations for which sensitivity analysis has been conducted.

Also, on L. 89 the authors say "This study simulates snow processes for the period November 2005 to June 2006. This choice was made to reduce the computational effort required for modelling and analysis".

The sentence suggests the model has long run time, but it is not clear to me where this complexity comes from if the model only simulate vertical fluxes – as suggested by Fig. 2 – at one location (the Aspen site). Or maybe the model does simulate the snow depth over the entire spatial domain, but then if so, how relevant it is to look at simulations (and their sensitivity) in one location only? This needs clarification.

**Response:** This is a valid question. We indeed simulated snow depth at the Aspen site over a defined spatial domain. Although the model runtime to generate the simulation results is not computationally intensive, conducting sensitivity analyses over the online PySUMMA platform requires a great deal of computational effort. To clarify, we used 20 trajectories (r) for each parameter (k), resulting in the number of simulations per parameter being r(k+1) = 20(1+1) = 40. With 12 parameters in total, this amounted to 40 * 12 = 480 simulations. To manage computational time effectively while covering a full snow season, we selected the period from November 2005 (before snowfall) to June 2006 (complete snow melt). We will update the methodology section to add this clarifying detail.

**3) PERFORMANCE METRICS**

In Table 2 and throughout the manuscript, the authors use the term "performance metric" to refer to all metrics, including the Mean (i.e. the average snow depth). However, differently from NSE, KGE and RMSLE, this statistic has nothing to do with the model performance (unless the Mean of simulated snow depth is compared to the mean of observations, which however does not seem to be the case from the equation in the last row of Table 2). This difference needs to be clarified. I'd suggest to use the term "performance metric" for NSE, KGE and RMSLE, and "output metric" for the Mean (this is often used in the Sensitivity Analysis literature for model outputs subject to SA which are based only on model simulations only).

**Response:** This is a good suggestion. Please refer to our response to comment #1.

The point should also be clarified in the results section.

On L. 247-248, the sentence "Overall, the performance results show a high degree of consistency between the simulated and observed snow depth data based on mean metric" is unclear. The consistency between simulated and observed data can be shown by the KGE, NSE and RMSLE metrics, but not the Mean metric. So, how did the authors get to that conclusion?

**Response:** We will rewrite this sentence to better reflect the objective of this paper which is the fact that performance metrics obscure the model behavior due to equifinality.

Similarly, on L. 270: "in cases where there is a discernible pattern in each parameter, it becomes possible to identify the optimum value that can lead to the highest level of agreement between the simulation and observation. For instance, the simulation of snow depth will improve for the values of the exponent for meltwater flow greater than 2" Again, how one can infer the level of agreement between the simulation and observation based on Figure 4, given it shows the (observations-free) mean of the simulated snow depth?

**Response:** I agree with the reviewer that using the metric "Mean" here does not accurately reflect the level of agreement between simulations and observations. We will thoroughly review our manuscript and remove the relevant descriptions. This change will not affect our conclusion regarding equifinality.

Given this comment, as we replied earlier, we will use two different terms in our revised manuscript: performance metric and behavior metric. The former compares simulations with observations, while the latter describes one of the system behaviors (without comparing to observations). Depending on the purpose of conducting sensitivity analysis, the adoption of these metrics can vary. If the goal is performance metrics-based model calibration (e.g., KGE, NSE, RMSLE), then performance metrics are appropriate. However, if the goal is to understand the sensitivity of system behavior (e.g., mean snow depth) or behavior metrics-based model calibration, then behavior metrics should be used.

Last, on L. 178 the authors mention "flow simulation results" but I believe the metrics used here are calculated on snow depth simulations results, not flows. Or are NSE, KGE and RMSLE computed over flow simulations? (but then if so, what flow observations are being used? And what about the other processes and parameters that determine the streamflow at gauging station?). This really needs to be clarified.

**Response:** That is correct. We actually meant "snow depth simulation results". This typo will be modified in the revised manuscript.

**MINOR**

L. 26 "that sensitivity analysis of snow modelling parameters plays a crucial role in understanding their impact on decision outcomes". This sentence is unclear (what "decision outcomes"?) please rephrase

**Response:** This will be rephrased.

L. 46 "interdisciplinary challenge" I would not say that "process-based hydrological modelling" is a particularly interdisciplinary work!

**Response:** Absolutely. This will be rephrased.

L. 53 "and different model parameters" suggest replacing "parameters" by "parameter values"

**Response:** The words "Parameters" and "Parametrizations" are used to reflect the difference between variables and equations, respectively. We will replace parameters by parameter values.

Figure 2 is unnecessarily large, consider reducing it. Also, using a consistent wording between this Figure and the first column of Table 1 would improve clarify (for example, I suppose "(partitioning)" in Fig. 2

refers to "Precipitation flux" in Table 1, "(Drainage parameterization" corresponds to "Liquid water in snowpack flux", etc.)

**Response:** Sure. The wording on the plot will be revised to match Table 1 in the revised version.

L. 179 "altering a parameter influences the outcome of a decision". Unclear, please rephrase.

**Response:** This will be rephrased.

Figure 4: units of measurements missing on vertical axis label

**Response:** The Unit (meter) will be added to the figure in the revised version.

L. 282: "The variance in the ranking of parameters' accuracy prediction by different performance metrics". This sentence does not make much sense, please revise.

**Response:** This will be rephrased.

L. 286: "is shown in Figure 6" should be "Figure 5" (I guess)

**Response:** This will be corrected in revised version.

L. 309: "the differences between the simulations with lowest thermal conductivity (blue line) have differences with the observations that are…" Convoluted/confusing sentence, please clarify.

**Response:** This will be rephrased.

Thank you once again for your valuable feedback. We believe our revisions will have significantly improve the clarity and robustness of our study. Please do not hesitate to contact us if you have any further questions or suggestions.

**References:**

McMillan HK. A review of hydrologic signatures and their applications. WIREs Water. 2021; 8:e1499. https://doi.org/10.1002/wat2.1499

Sincerely,
Tek Kshetri, Amir Khatibi, Yiwen Mok, Shahabul Alam, Hongli Liu, and Martyn P. Clark

---

## Author Comment (AC2)

**RC1: Steven Markstrom**

Thank you very much for your thorough review and insightful comments on our manuscript titled "Equifinality Contaminates the Sensitivity Analysis of Process-Based Snow Models." We appreciate your positive feedback on the organization and writing of our article, and we are grateful for your suggestions to improve our manuscript. Below, we address your general comments and minor suggestions in detail. Please find our response in **Blue** color.

**General Comments**

**Comment:** The word "equifinality" is in the title and it is an important concept to this article, but because this term has such a history in the literature, I think that a better description of what is specifically meant by equifinality is required. For example, is "parameter insensitivity" and "equifinality" the same within the context of this article?

**Response:** Thanks for the comment. Basically, we meant that the same parameters (e.g., the 12 parameters in Table 1) get different sensitivity rankings depending on the performance metrics used. To further clarify, no insensitivity concept was intended. We will add the following clarification to the revised manuscript.

**"Equifinality is a well-known concept in hydrological modeling that describes the situation where different combinations of model parameters can lead to similar model outputs (Beven, 2006). This phenomenon complicates the process of model calibration and sensitivity analysis because it can obscure the true relationships between parameters and model behavior. In the context of our study, equifinality refers to the ability of different parameter sets to yield similar performance metrics, which can mask the sensitivity of individual parameters. It is crucial to distinguish equifinality from parameter insensitivity, the latter being a condition where changes in parameters do not significantly affect model outputs. By clearly defining and addressing equifinality, we aim to enhance the reliability of sensitivity analyses in process-based snow models."**

**Comment:** Likewise, there needs to be more in the discussion about equifinality in the results section. For example, in the paragraph at lines 304 through 314, there is some discussion of "differences in model sensitivity" to different parameters and the identification that the model seems to be more sensitive to "water balance" parameters than "energy balance" parameters. What does this imply about model structure when some parameters exhibit equifinality and others don't? How could this information be used when modeling with SUMMA?

**Response:** These are valid and very constructive questions. It is believed that the higher sensitivity of certain process-based snow modelling parameters (e.g., water balance parameters and parametrizations) provide an information about how effective these parameters can be in the model calibration and/or an improvement to their physical descriptions can lead to higher accuracy of snow depth predictions. We will modify the discussion section to elaborate on the relations between parameters sensitivity and the model structure.

We would also like to note that, based on another reviewer's comment, we will make an important change to our discussion section by separating the "Performance Metrics" (i.e., RMSLE, KGE, NSE) from the "Signature Metric" (i.e., Mean). This change will subsequently impact the title and contribution of this study. Accordingly, less emphasis on equifinality in describing the model structure is expected; instead, the discussion will focus on the difference and application of these metrics (i.e., performance and signature metrics) on the model calibrations and behavior.

**Comment:** There is not mention of discretization and distribution of parameters and the simulated snowpack throughout space. Were the parameters varied spatially during the sensitivity analysis? Presumably the model simulates spatially varying snowpack (i.e., the snowpack is deeper in some locations than others) how did the objective function calculations account for this variability? Do the quality of the performance measures and their sensitivities vary according to how the depth of the pack vary across space? Maybe all of this is beyond the scope of the article, but some mention of how spatial variability is delt with is required.

**Response**: This is a good suggestion. We will revise the relevant part to the following in the revised manuscript.

**"**An important aspect not addressed in our current sensitivity analysis is the spatial variability of snowpack characteristics and parameter distributions. In this study, parameter sensitivity was analyzed using spatially averaged snow depth outputs in sheltered sites, which may not fully capture the spatial heterogeneity of snow processes. Future studies should incorporate spatially distributed parameters to better understand how spatial variability affects model sensitivity and performance. Acknowledging and addressing spatial variability will be critical in advancing the understanding of snow processes and improving model predictions."

**Specific comments:**

Ln. 221 This should reference figure 3 (not figure 4)
**Response:** This will be corrected in the revised manuscript.

Ln. 264 This should reference figure 4, (not figure 5)
**Response:** This will be corrected in the revised manuscript.

Ln. 286   This should reference figure 5 (not figure 6)
**Response:** This will be corrected in the revised manuscript.

Ln. 277 "of" instead of "as"
**Response:** This will be corrected in the revised manuscript.

Fig 4. Y axis label should be "Mean snow depth (m)".
**Response:** This will be corrected in ht erevised manuscript.

Thank you once again for your valuable feedback. We believe these revisions have significantly improved the clarity and robustness of our study. Please do not hesitate to contact us if you have any further questions or suggestions.

**References:**

Beven, K.: A manifesto for the equifinality thesis, J. Hydrol., 320, 18–36, https://doi.org/10.1016/j.jhydrol.2005.07.007, 2006.

Sincerely,
Tek Kshetri, Amir Khatibi, Yiwen Mok, Shahabul Alam, Hongli Liu, and Martyn P. Clark